

# Numerical Models for Monitoring and Forecasting Sea Level: a short description of present status

Angelique Melet[1], Begoña Pérez Gómez[2], Pascal Matte[3]

[1]Mercator Ocean International, Toulouse, France
[2]Puertos del Estado, Madrid, Spain
[3]Meteorological Research Division, Environment and Climate Change Canada, Québec, QC, Canada

*Correspondence to*: Angelique Melet (amelet@mercator-ocean.fr)

**Abstract.** Forecasting the sea level is crucial for supporting coastal management through early warning systems and for adopting adaptation strategies to climate changes impacts. Such objectives can be achieved by using advanced numerical models that are based on shallow water equations used to simulate storm surge generation and propagation due to atmospheric pressure and winds, or with ocean general circulation, baroclinic models. We provide here an overview on models commonly used for sea level forecasting, that can be based on storm surge models or ocean circulation ones, integrated on structured or
unstructured grids, including an outlook on new approaches based on ensemble methods.

## 1 Introduction

The low-elevation coastal zone, defined as the contiguous and hydrologically connected zone of land along the coast with an elevation above sea level of less than 10 m, covers only 2% of the world's land area but close to 10% of the world population lives there (Neumann et al., 2015). Due to the large economic value of coastal zones, economic losses due to coastal flood
risks induced by rising sea levels and extreme sea levels at the coast are huge (Abadie et al., 2020). Sea level rise and extremes can also exacerbate coastal erosion, saltwater intrusion and the degradation of coastal ecosystems.

A wealth of factors is influencing sea level changes at the coast (Woodworth et al., 2019). Extreme sea levels are due to the combination of different drivers: astronomical tides, storm surges, wind-waves setup and swash, mean sea level changes. Mean sea level changes are themselves induced by ocean circulation redistributing mass, heat and salt in the ocean, transfer of water
mass from land to the ocean (from mountain glaciers, ice-sheets, terrestrial water level storage changes). Mean sea level changes, including long-term trends, have been accurately monitored over the quasi-global ocean through satellite altimetry (Legeais et al., 2021). Sea levels at the coast, on the other hand, have been monitored thanks to tide gauges, whose data has been compiled in different datasets (e.g., Global Extreme Sea Level Analysis (GESLA3), Permanent Service for Mean Sea Level (PSMSL), Copernicus Marine Service). Tides, storm surges, and wind-waves can also change in response to climate
change (Haigh et al., 2019; Kirezci et al., 2020; Morim et al., 2019)

Numerical ocean models can be used to provide both consistent retrospective datasets of sea level changes over the global, regional, or coastal ocean, as well as forecasts of sea level change (Melet et al., 2021). Both can be used to support adaptation





to sea level rise (IOC-UNESCO, 2022). Due to sea level rise, the frequency of extreme sea levels at the coast will increase (Kirezci et al., 2020), and associated impacts on population and economic damages will too without further adaptation (Figure

1). Sea level short-term (a few days) forecasts provided by ocean forecasting systems are necessary information to feed early warning systems (EWS) for coastal floods. EWS are integrated systems allowing a real time monitoring of potential natural hazards, issuing warnings when a natural hazard is measured or forecasted, and informing stakeholders (e.g. civil protection agencies, regional and local authorities, ports, environmental agencies) as part of an integrated risk assessment cycle to mitigate risks. EWS were found to be an efficient adaptation measure by providing more than a tenfold return on investment (Global

Commission on Adaptation, 2019).

Monitoring of sea level change over past decades provides the historical baseline for quantifying sea level rise, extremes, their return periods and synoptic sea level variability in a broader sense. Ocean (wave) reanalyses combine ocean (wave) model dynamics with in situ and satellite observations through data assimilation. As such, reanalyses provide a consistent view of the ocean in space, time, and across variables, accounting for observation information and dynamics. The reliability of ocean

reanalyses has increased over the last decade (Forget et al., 2021; Lellouche et al., 2021; Storto et al., 2019; Zuo et al., 2019).

**2 Numerical models for forecasting sea level**

Numerical modeling systems are the backbone of ocean and wave hindcasts (modeling past evolutions over the last decades), reanalyses (hindcasts constrained by observations through routine assimilation of in situ and space observations) and forecasts (over a few days to weeks). Such models are solving the equations governing ocean and wave dynamics and are often

constrained by observations through assimilation of in situ and satellite observations (Alvarez-Fanjul et al., 2022). They provide a synoptic spatial and temporal monitoring of the ocean.

Regarding sea level forecasts, both storm surge models based on shallow water equations (Fujiang et al., 2022) and ocean general circulation models (OGCMs) based on primitive equations (Ciliberti et al., 2022) are used. In terms of model grids, both structured and unstructured grids can also be used. Other details on models equations, discretization methods, grid types,

coordinates, data assimilation techniques and inventory of operational systems. are available in Alvarez-Fanjul et al., 2022.

Wind-waves also contribute to mean and extreme sea levels through wave setup and to the fluctuation of the water line at the coast through wave runup (Dodet et al., 2019). Wind-wave sea level contributions are estimated from wave models (Aouf et al., 2022). In addition, non-linear interactions between mean sea level, tides, storm surges and waves are acting on the total sea level at the coast (Chaigneau et al., 2023; Idier et al., 2019).

The accuracy of numerical models to forecast sea levels are limited by the accuracy of the atmospheric forcing forecasts (especially so for the storm surge and wave components of total sea level changes at the coast), by tidal forcings for regional to coastal systems, the representation of bathymetry, by the lack of representation of non-linear interactions between sea level components (mean-sea level-tides-surges-wave), and by limitations of the ocean and wave models themselves.



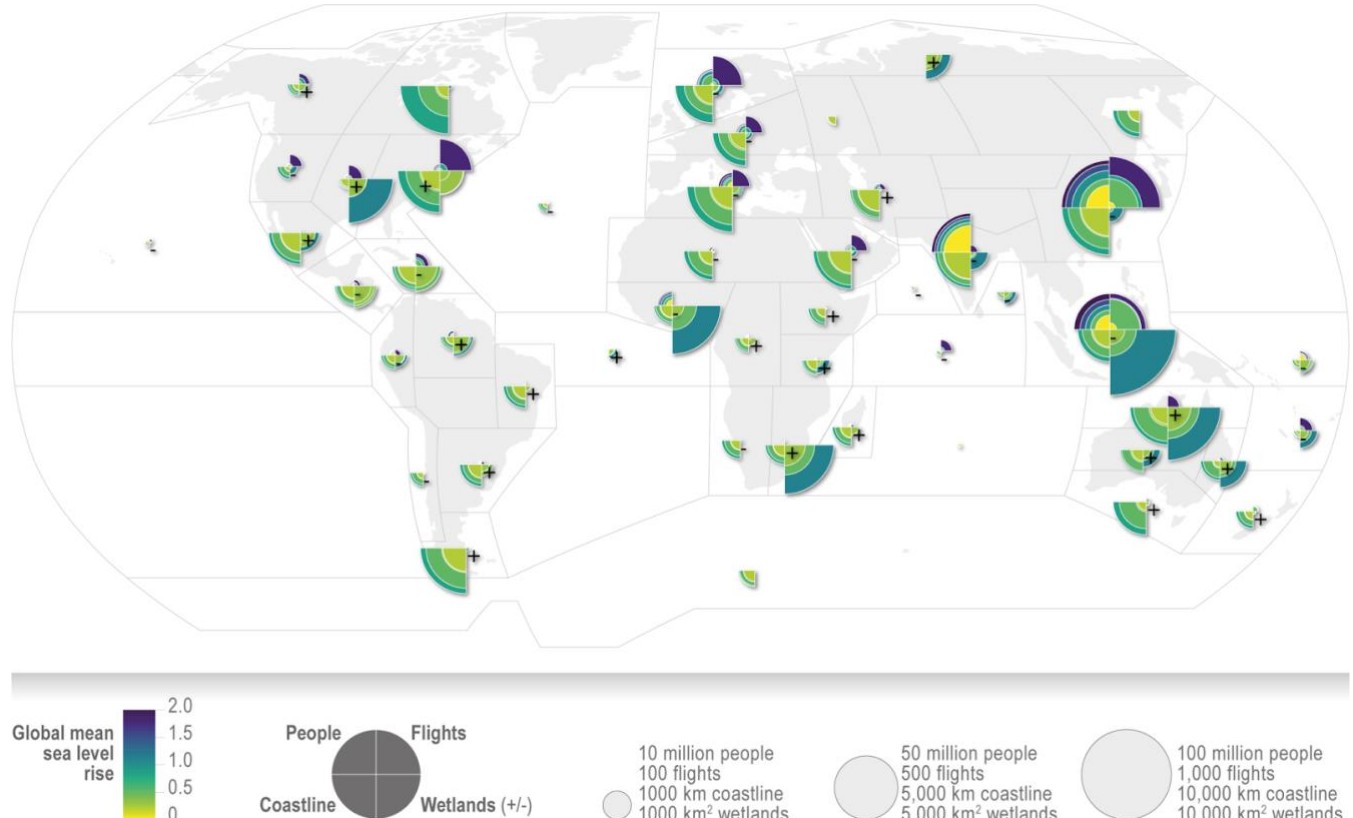

Figure 1: Map of risks for cities and settlements by the sea according to IPCC regions, extracted from IPCC AR6 (Glavovic et al., 2022). The map shows risks to people from a 100-year coastal flood event (*100.000)[16], risks to loss of coastal land (length of coast with more than 100 m retreat), risks to the built environment (airports at risk indicated by number of flights disrupted and risk to wetlands (± indicates positive or negative area change)[19]. Risks are reported against global mean sea level rise relative to 2020, depending on data availability.

## 3.1 Storm surge models

Storm surge models, also called hydrodynamic models here, are usually based on shallow water equations. They are the most common tools to simulate the generation and propagation of storm surges due to atmospheric surface pressure and winds, thereby providing water levels and velocities (e.g., SELFE, SCHISM, POM, Delft3D, ADCIRC, GTSM, MIKE21, TuFlow, ROMS, FVCOM) (Fujiang et al., 2022; Ciliberti et al., 2022). They can also incorporate astronomical tides. In these models, shallow water equations are often discretized based on unstructured meshes with either finite-volume methods or finite-element methods. Unstructured grids allow for a seamless modelling from the open to coastal ocean using a spatially variable resolution with finer resolution in the coastal zones (Figure 2). Mostly used in their 2D, barotropic version, such models are computationally-fast and can be used over continental-wide regions or the global ocean to produce hindcasts (Fernández-Montblanc et al., 2020; Fernández-Montblanc et al., 2019) reaching up to 1.25 km of resolution at the coast (Muis et al., 2020), operational forecasts (NOAA, 2023) and to produce tidal atlases (Lyard et al., 2021). However, barotropic hydrodynamic



models do not simulate changes in mean sea level, although this contribution can be substantial even for extreme sea levels in e.g., micro-tidal or non-stormy regions. 3D, baroclinic hydrodynamic models also exist. 3D baroclinic models are able to solve additional physical processes, such as the gradients of sea water density-induced changes in mean sea level (e.g., steric sea level), and lead to more accurate sea level and currents. Adding baroclinicity in a global barotropic operational model can lead

to significant improvements in predictions of extreme water levels (Wang et al., 2022) In storm surge models, the calibration of bottom friction is especially important. Such systems can assimilate different sources of observations notably to provide more accurate initial conditions for their forecasts and increase forecasts skills over short lead times. Observations assimilated in storm surge models include sea surface height from tide gauges, for higher frequency and coastal processes, and/or from satellite altimetry, for longer period processes. Operational storm surge forecasting systems have been implemented in many

countries, based on different types of storm surge models (Fujiang et al., 2022).

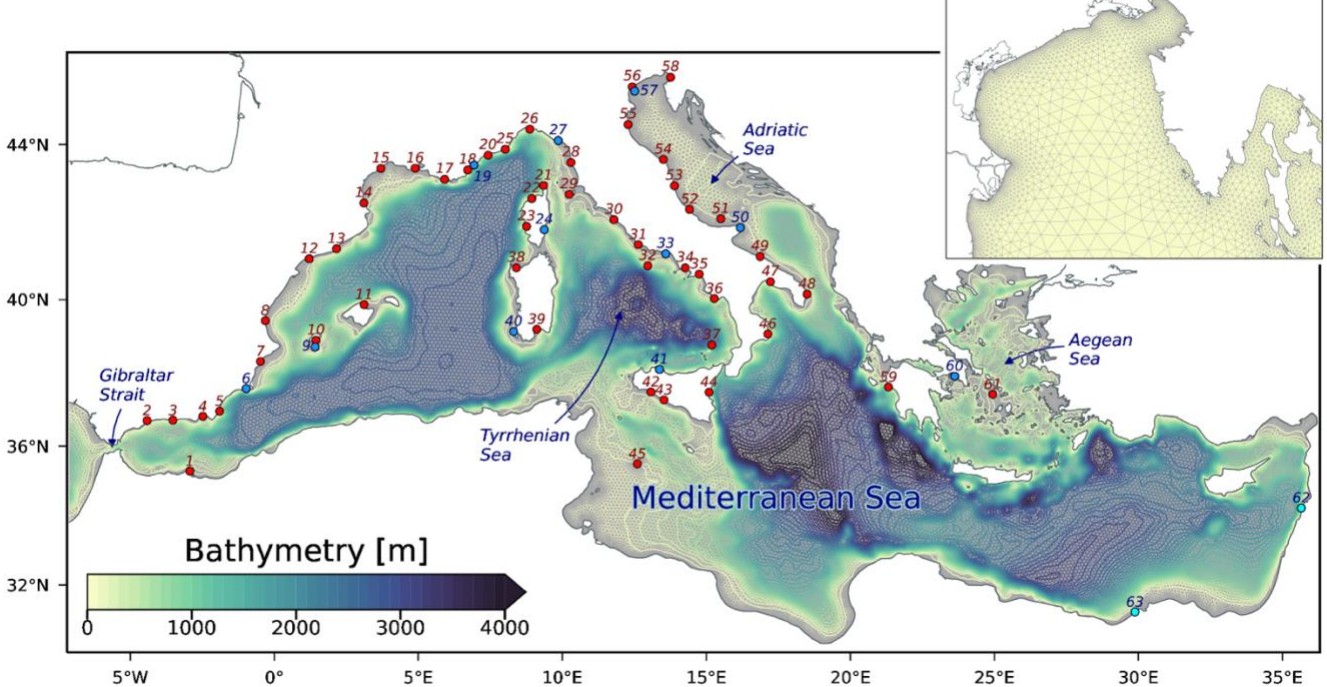

**Figure 2: An example of an unstructured barotropic ocean model and bathymetry used by the model (Bajo et al., 2023). The inset is a zoom of the grid in the northern Adriatic Sea.**

### 3.2 Ocean general circulation models

3D baroclinic ocean general circulation models, based on primitive equations (Ciliberti et al., 2022), are widely used in operational oceanography (e.g., NEMO, HYCOM, ROMS, MOM, MITgcm, CROCO, FVCOM, SHYFEM, SCHISM, FESOM, MPAS) for ocean circulation forecasting systems, also providing a valuable solution for forecasting sea level changes (Irazoqui Apecechea et al., 2023; Melet et al., 2021). More complex and expensive than storm surge models previously





described, they can simulate mean sea level changes due to ocean circulations as well as tides and storm surges when forced
by surface atmospheric pressure and wind, coherently with other ocean state variables (e.g., 3D temperature, salinity, ocean
currents). Operational systems also usually assimilate observations. Of particular importance for the representation of sea level
changes are the assimilation of satellite altimetry data, to directly constrain total sea level, in situ profiles of temperature and
salinity, to constrain the steric and dynamic component of sea level, and satellite gravimetry data, to constrain the mass
component of global mean sea level rise. The assimilation of satellite altimetry exerts a major constraint on such forecasting
systems to increase their skills (Hamon et al., 2019; Le Traon et al., 2017).

Due to the Boussinesq approximation in primitive-equation models, the global mean (or spatial average in an area-limited,
regional model) steric sea level change cannot be explicitly simulated. However, this time-dependent scalar can be diagnosed
from the temperature and salinity fields (Griffies and Greatbatch, 2012) and added to simulated sea level changes. Spatial
gradients of steric sea level changes are directly simulated in primitive equations models, through changes in temperature and
salinity inducing differences in density and circulation changes. Another limitation stems from the use of a constant, uniform
gravity field and the approximation of spherical geopotential surfaces. This approximation does not allow to represent the
changes in Earth gravity, rotation and solid-earth deformation (the so-called GRD effects) due to the transfer of water from
land to the ocean (e.g., melting mountain glaciers, mass loss of ice sheets, changes in land water storage), which contribute to
regional departures from the global mean sea level rise.

As hydrodynamical models, operational OGCMs can be used to forecast sea level changes from global scales (Global Ocean
Physics Analysis and Forecast, 2023) to coastal scales (Figure 3). For instance, the skills of the regional OOFS of the
Copernicus Marine Service covering European Seas to forecast sea level extremes were evaluated (Irazoqui Apecechea et al.,
2023) showing satisfactory performance, with yet an underprediction of peak magnitudes of both extreme sea levels and of
their surges component. For these OOFS, forecasts skills are stable for the first three days of the forecasts, but decreased at 4-
day and longer forecast lead times, demonstrating the suitability of the systems for early warning applications. Consideration
of the possible sea level processes included in these regional models must be made, when comparing/validating with local tide
gauge data. This may require additional pre-processing of tide gauge data to deal with higher frequency sea level oscillations
often recorded at very local scales and contributing to local extremes. Adding sea ice effects in a global operational model was
shown to improve total water level forecasts (Wang and Bernier, 2023).

Regional or global operational ocean forecasting systems can also be used to downscale sea level changes at more coastal
scales for local applications. Regional ocean models can have higher resolutions than global ocean models, (e.g., ranging from
2 km to 12 km for European Seas in the Copernicus Marine Service for operational forecasting systems as of July 2023),
benefit from ocean models adapted to the regional dynamics and from the representation of additional processes.





**Figure 3: Simulated (in dark red) and tide-gauge observed (in blue) sea-levels (including mean sea level, tides, surge) and surges during a selection of extreme events in Europe. (A)-Eleanor, 2018, Hoek Van Holland TG; (B)- Emma, 2018, Huelva TG; (C)-Vaia, 2018, Marina Di Campo TG; (D)-Detlef, 2019, Venice TG; (E)-Gloria, 2020, Valencia TG; (F)-Alfrida, 2019, Kiel TG. WL and surge percentile-thresholds for model and observations are shown with the corresponding colours, in horizontal dashed lines. Vertical blue line denotes the observed peak time for the plotted component. Extracted from Irazoqui Apecechea et al. 2023.**

Global and regional reanalysis can be used to provide a baseline over the past decades of sea level changes, when tide gauges are sparsely located along coastlines. Reanalyses benefiting from data assimilation capture the spatial variability of altimetry derived sea level trends[12]. Since altimetric observations capture sea level trends due to land ice mass loss and land water storage changes, in addition to trends due to sterodynamic sea level changes (Gregory et al., 2019), a processing of the altimetric data to be assimilated in OGCMs or a processing of the sea level represented in the model need to be performed. For instance, in the global ocean high-resolution reanalysis provided by the Copernicus Marine Service (GLORYS12, Lellouche et al., 2021), a global mean sea level trend is added at each time step to the modeled dynamic sea level, prior to data assimilation. This added GMSL signal is composed of the diagnosed global mean steric sea level change and of a barystatic (land ice related as in Gregory et al., 2019) sea level trend.



### 3.3 Ensemble forecasting

Deterministic solutions provided by numerical models can be complemented by multi-model systems, stochastic approaches and ensemble estimates. Ensemble forecasting allows to account for different sources of uncertainties that are arising from errors in, e.g., the initial or boundary conditions, the atmospheric forcing or forcing functions, the physics or parameterization of the numerical model, the bathymetry, the spatial or temporal resolution limitations. Forecast skills tend to decrease with increasing forecast lead times, as errors grow.. It is therefore possible to provide probabilistic forecasts that better support

coastal decision makers, by adding a confidence interval to the forecasted variable. This can be achieved in different ways (Alvarez-Fanjul et al., 2022), both for hindcasts and short-term forecasts, taking into account observational data to determine model performance and decrease model errors or not.

A first immediate approach is considering existing operational forecasts over an overlapping area, to build a multi-model system. This is today possible thanks to the number of general ocean circulation operational systems with a reliable coastal

sea level solution, as those of the Copernicus Marine Service (global and regional Marine Forecasting Systems MFCs). The good performance of these models for coastal sea level (Irazoqui Apecechea et al., 2023) can complement the solution provided by storm surge forecasting systems run at national level. This is the approach followed by Ports of Spain, which combines its 2D barotropic storm surge forecasting system (Nivmar, Alvarez-Fanjul et al., 2021), with the different MFC's covering the Spanish coast since 2012 (Pérez-Gómez et al., 2021). Today, the system, named ENSURF, combines Nivmar with two regional

MFCs, IBI-MFC (Aznar et al., 2016) and MedFS (Clementi et al., 2021). It makes use of the Bayesian Model Average (BMA) statistical technique (Beckers et al., 2008) for validation of the different models with tide gauge data in near-real time, and provides the outperforming mean and spread of sea level forecasts at the Spanish ports (Fujiang et al., 2022).

Thanks to the increased computational resources, storm surge ensemble forecasts can rely today on a larger number of members. A more recent multi-model and higher resolution approach is in place today for the Adriatic Sea, combining up to

19 sea level and waves models as described in (Ferrarin et al., 2020). Very often, the storm surge ensemble members are obtained by forcing the same model with an ensemble of meteorological forecasts providing different wind and sea level pressure fields, which account for most of the uncertainty during a storm. In this case, the model uncertainty will reflect the one of the meteorological forcing. As an example the ECMWF ensemble (Molteni et al., 1996) is used for storm surge operational forecasts in the North Sea (Flowerdew et al., 2010; Flowerdew et al., 2013). This approach has been applied also

for sea level forecasting in Venice by Mel and Lionello (2014).

Machine learning techniques can also be used to improve models performance locally, and account for high frequency sea level oscillations. This is the approach followed by Rus et al. (2023) in the Northern Adriatic, where traditional ensemble forecasting is replaced by computationally efficient machine-learning-based ensemble models, trained with tide gauge data to improve the probabilistic forecast and account for seiches at a single location.





## 4 Conclusions

Sea level forecasting is especially important at the coasts due to impacts on population and assets. Many operational systems are already in place, based on different model types, assimilating different observations (Capet et al., 2020; Fujiang et al., 2022; Ciliberti et al., 2022). Storm surge numerical modelling started in the 1950s, and operational oceanography with OGCMs combined with data assimilation largely developed in the 1980s and 1990s with the availability of satellite observations and increase in computational capacities. Despite decades of developments of such modeling systems and satisfactory forecast skills at short lead time, forecasting sea level changes at the coast at spatio-temporal scales relevant for decision-making remains challenging. This is notably due to the wealth of processes driving sea level changes at the coast and to the short scales of coastal zone dynamics.

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

**Competing interests**

The contact author has declared that none of the authors has any competing interests.