# Peer review of "Numerical Models for Monitoring and Forecasting Sea Level: a short description of present status"

_State of the Planet, 2024_

## Author Response (AR1)

**Numerical Models for Monitoring and Forecasting Sea Level: a short description of present status**

Angelique Melet[1], Begoña Pérez Gómez[2], Pascal Matte[3]

[1]Mercator Ocean International, Toulouse, France
[2]Puertos del Estado, Madrid, Spain
[3]Meteorological Research Division, Environment and Climate Change Canada, Québec, QC, Canada

*Correspondence to*: Angelique Melet (amelet@mercator-ocean.fr)

**Reviewer 1- Anonymous**

The authors propose an overview of the numerical models currently available to support the monitoring and forecasting of sea level variability. In the context of climate change, various factors drive changes in the observed mean sea level and the occurrence of extreme sea level events. The manuscript summarizes the characteristics and limitations of these numerical models, emphasizing their relevance in supporting adaptation strategies and early warning systems.

The paper is well-written and organized, with a clear objective outlined in the title and abstract. The proposed overview is valuable for readers seeking a general introduction to the current state of numerical models supporting the sea level analysis. However, in my opinion, the intended target audience of this manuscript remains unclear. Readers already familiar with this topic might benefit from additional references to key scientific results, while a general audience needs some more clarifications on the differences between the models (see specific comments).

The expected audience are scientist with an interest on Ocean forecasting, but not necessarily experts on the sea level field.

Considering the identified weaknesses, I believe that the paper is suitable for publication after undergoing a minor revision process.

We thank the reviewer for their constructive criticism and time spent on the manuscript. We address your comments below:

**Specific comments:**

In general, most of the information is supported by references, but some sentences are not, making it difficult for readers to explore further if interested. I found two main points:

- to support the description of unstructured meshes (section 3.1). Although it is a detail in the model description, their implementation significantly enhances the simulation of coastal processes.

  We added a few references in section 2.1 (renumbered from 3.1) – Storm surge models to highlight this point:

"Unstructured grids allow for a seamless modelling from the open to coastal ocean using a spatially variable resolution with finer resolution in the coastal zones (Figure 1), which enhances the simulation of coastal processes (e.g., Federico et al., 2017; Ferrarin et al., 2018; Toomey et al., 2022; Zhang et al., 2016)."

Zhang, Y. J., Stanev, E., and Grashorn, S.: Unstructured-grid model for the North Sea and Baltic Sea: validation against observations. Ocean Model., 97, 91-108. https://doi.org/10.1016/j.ocemod.2015.11.009, 2016.

Federico, I., Pinardi, N., Coppini, G., Oddo, P., Lecci, R., and Mossa, M.: Coastal ocean forecasting with an unstructured grid model in the southern Adriatic and northern Ionian seas. Nat. Hazards Earth Syst. Sci., 17, 45-59. https://doi.org/10.5194/nhess-17-45-2017, 2017.

Ferrarin, C., Bellafiore, D., Sannino, G., Bajo, M., and Umgiesser, G.: Tidal dynamics in the inter-connected Mediterranean, Marmara, Black and Azov seas. Prog. Oceanogr., 161, 102-115. https://doi.org/10.1016/j.pocean.2018.02.006, 2018.

Toomey, T., Amores, A., Marcos, M., Orfila, A.: Coastal sea levels and wind-waves in the Mediterranean Sea since 1950 from a high-resolution ocean reanalysis. Front. Mar. Sci. 9:991504. doi: 10.3389/fmars.2022.991504, 2022.

- to describe the GRD effects, as these are usually not considered by ocean models.

References were added in section 2.2:

"This approximation does not allow to represent the changes in Earth gravity, rotation and solid-earth deformation
(the so-called GRD effects, Gregory et al., 2019, Mitrovica et al., 2011) due to the transfer of water from land to the ocean (e.g., melting mountain glaciers, mass loss of ice sheets, changes in land water storage), which contribute to regional departures from the global mean sea level rise."

Mitrovica, J. X., Gomez, N., Morrow, E., Hay, C., Latychev, K., and Tamisiea, M. E.: On the robustness of predictions of sea level fingerprints: On predictions of sea-level fingerprints, Geophysical Journal International, 187, 729–742,
https://doi.org/10.1111/j.1365-246X.2011.05090.x, 2011.

I also suggest enhancing the description of the factors that limit the model accuracy as a key point in Section 2.

The paragraph has been slightly developed (to avoid lengthening the paper) and reads:

"The accuracy of numerical models to forecast sea levels is limited by several factors (e.g., discussion in Irazoqui Apecechea et al., 2023), such as the accuracy of the atmospheric forcing forecasts (especially so for the storm surge and wave components
of total sea level changes at the coast), by tidal forcings for regional to coastal systems, by the representation of bathymetry, by the lack of representation of non-linear interactions between sea level components (mean-sea level-tides-surges-wave), and by limitations of the ocean and wave models themselves (e.g., model numerics, resolution, lack of some coastal processes such as wetting and drying, river-estuary-ocean continuum)."

Finally, I would clarify the differences between 3D hydrodynamic models and 3D baroclinic OGCMs (sections 3.1 and 3.2).

When models are used in their baroclinic version, the main difference between hydrodynamic / "storm surge" models and OGCMs relates to the grid (unstructured vs structured) and induced differences in exchanges between cells, as well as the use of shallow water equation in most storm surge models vs primitive equations in OGCMs, as stated in sections 3.1 and 3.2. Since these elements are already in sections 3.1 and 3.2, we did not update the text in that regard.

**Technical corrections:**

Some acronymous are not defined in the text, like TG, WL, OOFS, MFC.

Thanks for noting it. Acronyms were expanded in the text (TG: tide gauges, WL: water level, OOFS: operational ocean forecasting systems, MFC: monitoring and forecasting center). The WL acronym is not used anymore, MFC was already defined (line 156).

Figure captions could be improved to allow readers to understand the content without looking at the original source.

In Figure 1: the global mean sea level rise units are not specified (bottom-left colorbar), and the references in numbers have no correspondence within the manuscript.

Thanks. Units for global mean sea level rise in this figure is meters. References in this caption are proper to the datasets used to produce this figure, and are now explicit (author, year) and not indexed as numbers.

The caption has been modified to stick more closely to the caption of the IPCC chapter (Glavovic et al., 2022):

Figure 1: Map of risks for cities and settlements by the sea according to IPCC regions, extracted from IPCC AR6 (Glavovic et al., 2022). The map shows risks to people (number of people at risk from a 100-year coastal flood event; Haasnoot et al., 2021), risks of loss of coastal land (length of coast with more than 100 m retreat; Vousdoukas et al., 2020), risks to the built environment (airports at risk indicated by expected annual number of flights disrupted by coastal flooding (Yesudian and Dawson, 2021) and risk to wetlands (± indicates positive or negative area change; Schuerch et al., 2018). Risks are reported against global mean sea level rise relative to 2020 (in meters), depending on data availability.

Haasnoot, M., Winter, G., Brown, S., Dawson, R. J., Ward, P. J., and Eilander, D.: Long-term sea-level rise necessitates a commitment to adaptation: A first order assessment, Climate Risk Management, 34, 100355, https://doi.org/10.1016/j.crm.2021.100355, 2021.

Vousdoukas, M. I., Ranasinghe, R., Mentaschi, L., Plomaritis, T. A., Athanasiou, P., Luijendijk, A., and Feyen, L.: Sandy
coastlines under threat of erosion, Nature Climate Change, 10, 260–263, https://doi.org/10.1038/s41558-020-0697-0, 2020.

Yesudian, A. N. and Dawson, R. J.: Global analysis of sea level rise risk to airports, Climate Risk Management, 31, 100266, https://doi.org/10.1016/j.crm.2020.100266, 2021.

Schuerch, M., Spencer, T., Temmerman, S., Kirwan, M. L., Wolff, C., Lincke, D., McOwen, C. J., Pickering, M. D., Reef, R.,
Vafeidis, A. T., Hinkel, J., Nicholls, R. J., and Brown, S.: Future response of global coastal wetlands to sea-level rise, Nature, 561, 231–234, https://doi.org/10.1038/s41586-018-0476-5, 2018.

In Figure 2: for clarity, I would specify which model uses the presented grid and what the red and blue dots refer to. The tide gauges network is not directly described in the text, but it is mentioned in line 88 to improve storm surge models.

The caption has been updated to:

**Figure 1: An example of an unstructured barotropic ocean model and bathymetry (here, from the SHYFEM -System of HydrodYnamic Finite Element Modules- model; Bajo et al., 2023). The inset is a zoom of the grid in the northern Adriatic Sea. The blue and red dots mark the locations of tide gauges.**

As the figure is used for illustration purpose, we don't think we need to discriminate between the blue and red dots in the caption (noting that the red and blue dots mark the locations of the assimilation and validation tide gauges for this simulation
respectively).

In Figure 3: the acronymous are not defined (TG and WL), making the figure difficult to understand for a general reader.
Acronyms were expanded.

Typing errors:

Line 33: "IOC-UNESCO, 2022" is not in the references list. Is it the same of Aouf et al (2022) ?

The reference is:

Alvarez Fanjul, E., Ciliberti, S., Bahurel, P.: Implementing Operational Ocean Monitoring and Forecasting Systems. IOC-UNESCO, GOOS-275. https://doi.org/10.48670/ETOOFS, 2022.

It has been corrected in the main text. The reference is already in the list of references.

Line 55: "operational systems. Are available"

Corrected.

Line 67: a closing parenthesis is missing in "number of flights disrupted and risk to"

Corrected.

Line 137: change reference style for "[12]"

Done. The reference is :

Lellouche, J.-M., Greiner, E., Bourdallé-Badie, R., Garric, G., Melet, A, Drevillon, M., Bricaud, C., Hamon, M., Le Galloudec, O., Regnier, C., Candela, T., Testut, C.-E., Gasparin, F., Ruggiero,, G., Benkiran, M., Drillet, Y., and Le Traon, P. Y.: The Copernicus global 1/12° oceanic and sea ice GLORYS12 reanalysis. *Frontiers in Earth Science*, 9, 698876, https://dx.doi.org/10.3389/feart.2021.698876, 2021.

Line 149: double period ".."

Corrected.

Line 158: typo in the reference year of "Alvarez-Fanjul et al., 2021" (2001?)

The reference is related to the NIVMAR system. The correct year is indeed 2001. It has been corrected in the text.

Alvarez-Fanjul, E., Pérez-Gomez, B., and Rodríguez, I.: Nivmar: a storm surge forecasting system for Spanish waters. Scientia Marina. Vol. 65, pp. 145–154. https://doi.org/10.3989/scimar.2001.65s1145, 2001.

Line 217: the publication year is missing in Dodet et al. (2019?)

Yes, this has been corrected.

**Reviewer 2- Georg Umgiesser**

General comments:

The article gives an overview on sea level forecasting. It does not go much into much detail, simply showing the applications and inherent problems of modelling the sea level, both for storm surge applications and for general circulation applications. Most of the available models are cited, but not referenced, but this would probably be overkill for such a short article as this one.

I have some slight changes to suggest before the article can be published. Therefore, the article can be published after little minor review.

**We thank the reviewer for his review, for the time spent and constructive criticism.**

Specific comments:

70: there needs to be a Section Title "3. Numerical Models" with some text to explain what comes next, that you divided the discussion into storm surge, GCM, and ensemble forecast models. Then you can start with 3.1

**Subsections 3.1, 3.2 and 3.3 actually belong to section 2 "Numerical models for forecasting sea level". They were corrected to 2.1, 2.2 and 2.3. Thanks for noting this.**

73: since you make reference to figure 2 some lines below, which comes from an article where the authors have used SHYFEM, it would be just logical to include into the list also SHYFEM as a storm surge model.
**An explicit mention to SHYFEM has been added to the caption of Figure 2, as requested by Reviewer 1.**

**The caption of Figure 2 has been updated to: Figure 2: An example of an unstructured barotropic ocean model and bathymetry (here, from the SHYFEM -System of HydrodYnamic Finite Element Modules- model; Bajo et al., 2023). The inset is a zoom of the grid in the northern Adriatic Sea. The blue and red dots mark the locations of tide gauges.**

**We also added SHYFEM in the non-exhaustive list of storm surge models in section 2.1.**

81: "do not simulate changes in mean sea level…" please be more precise. They do not simulate sea level anomalies due to baroclinic effects.

**We developed the sentence to :**

**"However, barotropic hydrodynamic models do not simulate changes in mean sea level due to baroclinic effects (…)"**

85: missing full stop after (Wang et al., 2022)

**Thanks, the full stop has been added.**

Figures: as I can see it, the figures have been reproduced from existing articles, which have been also referenced. However, these references should be inserted in the Reference list at the end of the article. As I can see, the reference to Bajo et al., 2023 is missing.

**References cited in captions are in the list of reference, including Bajo et al. 2023.**